# The Epidemiology of Mobility Difficulty in Saudi Arabia: National Estimates, Severity Levels, and Sociodemographic Differentials

**DOI:** 10.3390/healthcare13151804

**Published:** 2025-07-25

**Authors:** Ahmed Alduais, Hind Alfadda, Hessah Saad Alarifi

**Affiliations:** 1Department of Psychology, Norwegian University of Science and Technology, NO-7491 Trondheim, Norway; 2Department of Curriculum and Instruction, College of Education, King Saud University, Riyadh 11362, Saudi Arabia; 3Department of Educational Administration, College of Education, King Saud University, Riyadh 11362, Saudi Arabia; arifi-hs@ksu.edu.sa

**Keywords:** mobility difficulty, physical difficulty, prevalence, assistive devices, Saudi Arabia

## Abstract

**Background:** Mobility limitation is a pivotal but under-documented dimension of disability in Saudi Arabia. Leveraging the 2017 National Disability Survey, this cross-sectional study provides a population-wide profile of mobility-related physical difficulty. **Objectives:** Five research aims were pursued: (1) estimate national prevalence and severity by sex; (2) map regional differentials; (3) examine educational and marital correlates; (4) characterize cause, duration, and familial context among those with multiple limitations; and (5) describe patterns of assistive-aid and social-service use. **Methods:** Publicly available aggregate data covering 20,408,362 Saudi citizens were cleaned and analyzed across 14 mobility indicators and three baseline files. Prevalence ratios and χ^2^ tests assessed associations. **Results:** Overall, 1,445,723 Saudis (7.1%) reported at least one functional difficulty; 833,136 (4.1%) had mobility difficulty, of whom 305,867 (36.7%) had mobility-only impairment. Severity was chiefly mild (35% of cases), with moderate (16%) and severe (7%) forms forming a descending pyramid. Prevalence varied more than threefold across the thirteen regions, peaking in Aseer (9.4%) and bottoming in Najran (2.9%). Mobility difficulty clustered among adults with no schooling (36.1%) and widowed status (18.5%), with sharper female disadvantage in both domains (*p* < 0.001). Among those with additional limitations, chronic disease dominated etiology (56.3%), and 90.1% had lived with disability for ≥25 years; women were overrepresented in the longest-duration band. Aid utilization was led by crutches (47.7%), personal assistance (25.3%), and wheelchairs (22.6%), while 83.8% accessed Ministry rehabilitation services, yet fewer than 4% used home or daycare support. **Conclusions:** These findings highlight sizeable, regionally concentrated, and gender-patterned mobility burdens, underscoring the need for education-sensitive prevention, chronic-care management, investment in advanced assistive technology, and distributed community services to achieve Vision 2030 inclusion goals.

## 1. Introduction

### 1.1. Defining Mobility Difficulty and Activity Limitations

Mobility difficulty—often framed as an activity limitation—covers challenges that hinder changing or maintaining body position, handling objects, walking, or using transport. The International Classification of Functioning groups these actions into core domains of posture change, object manipulation, and locomotion [1]. Contemporary bedside tools such as the icon-based ProMover scale translate those domains into practical items—sit-to-stand transfers, stair climbing, and ambulation with or without aids—allowing rapid clinical quantification [2]. Beyond biomechanics, qualitative work shows that older adults regard mobility loss as a multidimensional burden that curtails social participation and erodes autonomy [3]. Clinically, mobility limitation clusters with geriatric risk factors—aging, inactivity, obesity, and multimorbidity—each amplifying incident impairment [4,5]. Consequences cascade to frailty, falls, and cognition; longitudinal evidence links deteriorating gait to accelerated cognitive decline [6]. Targeted exercise—resistance, balance, and gait training—mitigates these risks and enhances quality of life [7,8], yet accelerometer studies reveal many severely limited adults achieve fewer than 3000 daily steps, underscoring a gap between prescription and lived behavior [9]. Collectively, this constellation establishes mobility difficulty as a priority for epidemiologic surveillance and intervention.

### 1.2. Prevalence of Mobility Difficulties and Activity Limitations

Mobility limitation is neither uncommon nor restricted to old age. In U.S. adults aged 50–64, more than one-third report functional constraints, and 11% already struggle to walk a quarter mile or climb ten steps [10]. Comparable self-reports in Estonia show 18.5% of adults aged 20–79 with activity restrictions, one in ten of which are severe [11]. Finnish data add a performance lens: one-third of women and one-fifth of men aged ≥55 cannot reach a 1.2 m/s gait speed, a community-safety benchmark, and indoor mobility problems surge after age 75 [12]. Even “well-functioning” adults aged 70–79 display latent endurance deficits—23% of men and 36% of women fail objective walk tests despite denying problems [13]. Prevalence also tracks social stratification. A decade-long U.S. panel found non-Hispanic Black adults ≥60 with the highest burden (33%) versus Whites (29%) and Hispanics (26%) after adjustment [14]. In Singapore, low education, widowhood, and low income elevate limitation risk [15]. Pediatric evidence from the Netherlands shows 35% of children with spastic cerebral palsy needing mobility assistance [16]. Forecasts for Finland predict a doubling of severe mobility limitation by 2044, powered by aging, excess weight, and inactivity [17]. Together, these studies depict a widespread, socially patterned, and rapidly expanding burden that demands timely preventive and rehabilitative strategies.

### 1.3. Classification Frameworks for Mobility Difficulty

The World Health Organization’s International Classification of Functioning, Disability and Health (ICF) provides the most granular taxonomy for mobility limitation. Within its “Activities and Participation” domain, codes d410–d469 address changing or maintaining body position, carrying or handling objects, and walking or moving, while d470–d489 cover transport use and other or unspecified mobility [18,19]. These codes translate observable actions into functional capacity, giving clinicians, researchers, and policymakers a shared language [20]. Empirical validation is strong: the 22-item Mobility Activities Measure aligns with ICF sub-domains and loads onto five expected factors—two posture-change, two object-handling, and one walking [1]. Digital tools such as EasyICF embed these codes in electronic records, enhancing interdisciplinary assessment [21]. Sports-therapy scholars likewise cite ICF mobility (d4) as a “central health criterion,” facilitating translation of exercise goals into health-system metrics [22].

ICF pairs naturally with the International Classification of Diseases (ICD-11), whose disease-specific etiological codes link to mobility codes to create full biopsychosocial profiles [23]. This dual framework underpins global surveillance and administrative datasets, promoting comparability of mobility statistics [24]. By contrast, the Diagnostic and Statistical Manual of Mental Disorders (DSM-5) centers on psychiatric syndromes; although it now includes functional-impairment qualifiers, it lacks mobility-specific activity codes [25,26]. Critics note DSM’s narrower impairment lens cannot capture the full participation spectrum described by ICF [27]. In sum, ICF supplies detailed activity codes, ICD frames underlying conditions, and DSM contextualizes comorbid mental health—an integrated toolkit that informs subsequent discussions of risk factors and intervention strategies.

### 1.4. Theoretical and Conceptual Frameworks for Understanding Mobility Difficulty

Foundational disablement theorists distinguished bodily impairment from the activity limitations and participation restrictions it triggers. Nagi’s model, later expanded by Verbrugge, traces a progression from pathology to disability shaped by personal and environmental influences [15]. The ICF formalizes this biopsychosocial chain, mapping tasks such as changing body position, walking, and using transport to broader activity and participation constructs; qualitative and survey evidence confirm its reach, revealing how older adults adapt tasks and how ICF mobility codes mirror chronic disease, mental health, and socioeconomic disparities [28,29,30]. Building on these roots, Webber, Porter, and Menec propose five intersecting determinant clusters—physical, cognitive, psychosocial, environmental, and financial—filtered by gender, culture, and biography, paralleling social-ecological models that locate mobility within multilevel contexts [31,32,33]. Adaptive theories such as Selection, Optimization, and Compensation show individuals reshaping routines to preserve autonomy under long-term impairment [34]. Clinical syntheses underscore age, inactivity, obesity, and multimorbidity as modifiable levers and endorse resistance-balance training to bolster mobility [4]. Environmentally, sidewalk network connectivity governs active travel [35], while sociological analyses frame walking as a socially organized practice embedded in cultural meaning [36]. Collectively, these layered frameworks portray mobility difficulty as a multifactorial phenomenon, guiding subsequent discussions of modifiable risks and interventions.

### 1.5. Mobility Difficulties and Activity Limitations in Saudi Arabia

In Saudi Arabia, mobility challenges stem from exceptionally low physical activity intersecting with rapid, car-oriented urbanization. A national survey of 17,395 adults aged 30–70 revealed 96% were inactive—98% of women and 94% of men—with inactivity highest in the central region [37]. Qualitative and survey work in Jeddah attributes this to extreme heat, dispersed land use, scant pedestrian infrastructure, and gendered social norms; women cite societal restrictions, whereas men mention clothing and status concerns [38,39]. Consequently, functional limits span the ICF spectrum: in youth, higher body-mass index predicts difficulty walking or running, curbing school and social participation [40], while Riyadh data show boys make more walking trips than girls, embedding early mobility inequality [41]. Among adults with existing impairment, one-third of community-dwelling older people report disability, and poor 30-Second Chair Stand and low balance confidence greatly increase limitation odds [42]. Although smart wheelchairs offer navigational assistance, users cite high costs and maintenance barriers [43]; basic aids still dominate, prompting calls for user-centered frameworks such as PHAATE [44]. Rehabilitation priorities among neurological inpatients center on functional mobility, with goal attainment linked to self-reported difficulty [45], and family cohesion predicts better recovery after trauma [46]. Vision 2030 champions active living, and pandemic analyses show that measures such as a one-kilometer walking limit plus well-distributed neighborhood parks can improve access to activity space [47]. These studies depict a landscape where sociocultural norms, urban design, health services, and assistive technology jointly shape mobility limitations, signaling multiple leverage points for policy and practice.

### 1.6. Purpose of the Present Study

Despite a rich body of global work describing the conceptual foundations of mobility limitation (e.g., Nagi’s disablement model as cited in [15]; the ICF framework as cited in [18,19]), its epidemiology across age groups and multifactorial determinants, empirical evidence for Saudi Arabia remains fragmentary and disjointed. International studies consistently document high and rising prevalence of mobility difficulty, strong links with chronic conditions, sex and socioeconomic gradients, and effectiveness of exercise-based interventions; yet Saudi data are largely confined to isolated reports of extreme physical inactivity, survey snapshots of assistive-device use, or qualitative accounts of cultural barriers to walking. Theoretical frameworks such as Webber’s comprehensive mobility model and the ICF emphasize the interplay of individual, social, and environmental factors, but no national analysis has simultaneously mapped prevalence, severity, sociodemographic correlates, familial context, etiology, and service utilization within a single Saudi cohort. Moreover, previous Saudi studies seldom disaggregate mobility difficulty by administrative region, overlook education and marital status as structural drivers, and rarely examine how sex moderates patterns of multiple disabilities, aid uptake, or ministry service use. This evidentiary gap limits the Kingdom’s ability to benchmark its mobility-related disability burden against international norms, to target high-burden regions and vulnerable subpopulations, and to align rehabilitation and social-service planning with the nuanced realities of those affected.

Based on this gap, the present study addresses five specific questions about the epidemiology of mobility difficulty in Saudi Arabia: (RQ1) what is the national prevalence and severity distribution of mobility-related physical difficulty among Saudi citizens, and how do these estimates differ by sex?; (RQ2) how does the prevalence of mobility difficulty vary across the thirteen administrative regions of Saudi Arabia, and which regions exhibit significantly higher or lower burdens after accounting for population size?; (RQ3) to what extent are sociodemographic factors—specifically educational attainment and marital status—associated with mobility difficulty, and do these associations differ between men and women?; (RQ4) what familial relationships, proximate causes, and duration profiles characterize Saudis who experience mobility difficulty in combination with at least one additional functional limitation, and how do these contextual factors differ by sex?; and (RQ5) what types of mobility aids and social development services are most frequently used by Saudis with mobility difficulty, and are there statistically significant sex differences in utilization patterns?

## 2. Methods

### 2.1. Design

The current study employed a descriptive cross-sectional research design to examine the prevalence and characteristics of mobility-related physical difficulties among the Saudi population.

### 2.2. Sample

The sample for this study was drawn from the Disability Survey 2017 conducted by the General Authority for Statistics (GAStat), Saudi Arabia. This survey employed a two-stage random categorical cluster sampling method designed to produce efficient and reliable national estimates. The sampling framework was derived from the updated population and housing census of 1431H (2010), revised in 2016. The sample design involved dividing the Saudi society into non-overlapping, relatively homogeneous clusters or categories. Each category was treated as an independent subpopulation from which random samples were independently drawn [48]. In the first stage, 1344 primary sampling units (PSUs) were selected from 3600 statistical areas distributed proportionally across the 13 administrative regions, including Riyadh (217 PSUs), Makkah (242), Eastern Region (168), and smaller allocations to other regions based on population size [48]. In the second stage, a systematic random sampling method selected up to 25 households from each PSU, yielding a total of 33,575 households nationally. Regional household samples ranged from 6050 in Makkah to 1500 in Al-Bahah, ensuring adequate representation and statistical precision at both national and regional levels [48].

The survey covered all Saudi households residing within the Kingdom, including Saudi nationals temporarily abroad for purposes such as treatment, study, or tourism, provided they remained household members. The inclusion criteria for household members encompassed all individuals normally residing with the household during enumeration, including those temporarily absent due to work shifts, business travel, or studies [48]. Such rigorous sampling and comprehensive inclusion criteria ensured the representativeness and generalizability of the results. This meticulous sampling strategy aligns with international standards recommended by organizations such as the Washington Group on Disability Statistics, enhancing the survey’s comparability with global disability data [49,50].

Regarding exclusion, the GAStat defines the survey universe as Saudi households inside the Kingdom and Saudis who are temporarily abroad for treatment, study, tourism, etc., so long as they remain members of a sampled household. All 13 administrative regions were included. No other eligibility or exclusion criteria are specified; consequently, units lying outside that universe (e.g., non-Saudi households or institutional group quarters not counted as private Saudi households) were not part of the sampling frame by definition.

### 2.3. Measures

This study utilized secondary data derived from the 2017 Disability Survey conducted by the GAStat, Saudi Arabia. The survey employed a structured questionnaire developed following international guidelines, specifically the extended set of disability questions (Washington Group—Extended Set Questionnaire “ES-Q”). The ES-Q is recognized globally for its ability to produce comparable disability data by capturing information on various types and levels of functional difficulties, including mobility [51,52]. In this survey, mobility difficulty was explicitly defined as experiencing significant limitations in activities involving walking or climbing stairs. Respondents were classified into three severity levels based on their reported ability: mild (difficulty walking or climbing stairs but able with effort), severe (significant difficulty walking or climbing stairs requiring substantial assistance), and extreme (completely unable to walk or climb stairs independently). Such classification aligns with the Washington Group’s established method for measuring disability, which categorizes difficulties by the intensity of the functional limitation [49,52].

To fully analyze mobility difficulty, we specifically selected and used 14 relevant indicators from the survey dataset, grouped into four broad dimensions: (1) prevalence and severity of mobility difficulty, (2) sociodemographic characteristics (education and marital status), (3) contextual factors (parental relationship, cause, and duration), and (4) use of mobility aids and supportive services. These indicators provided an integrated view of mobility difficulties and enabled detailed profiling and comparative analyses across sex and regional categories.

The Washington Group Extended Set on Functioning (WG-ES)—the instrument on which the 2017 Saudi Disability Survey is based—contains 34 mandatory questions plus 3 optional items (total = 37) that span ten functional domains, including vision, hearing, mobility, cognition, self-care, upper-body use, communication, affect, pain, and fatigue. Of this total, the mobility domain comprises eight specific items (MOB_1—MOB_8). These cover general walking/climbing difficulty, use of mobility aids, detailed distance and stair-climbing capacities both with and without aids, and the types of equipment or personal assistance employed (see Appendix A).

### 2.4. Procedure

#### 2.4.1. Data Collection and Preparation

The data utilized for this study were publicly available secondary data obtained from the GAStat, Kingdom of Saudi Arabia. The survey data, openly published on the GAStat official website, was downloaded in the form of Excel spreadsheets containing aggregated counts and percentages for the entire Saudi population by region, sex, severity, and other relevant variables. Fourteen indicators specifically related to mobility difficulty—including severity, sociodemographic characteristics, parental relationships, causes, duration, multiple disabilities, mobility aid utilization, and service use—were carefully identified and extracted. Additionally, three baseline datasets provided population-level data (total population, age, and nationality). Data cleaning involved verifying consistency across indicators, resolving minor discrepancies, clarifying definitions, and standardizing column headers to ensure precise and consistent analyses. This data cleaning and organization process was necessary to facilitate accurate prevalence calculations and descriptive analyses, aligning with best practices in secondary data analysis [53].

#### 2.4.2. Data Analysis

Data analysis primarily employed descriptive statistics and chi-square tests to evaluate associations between categorical variables. Prevalence rates were calculated by dividing the number of individuals reporting mobility difficulties by the total relevant population, with severity distributions presented as proportions. Chi-square tests (χ^2^) were applied to assess differences across demographic groups (e.g., sex and region) and to determine statistical significance in the distribution of key variables such as severity, cause, marital status, parental relationships, duration of difficulty, and use of mobility aids and supportive services. Chi-square tests were chosen because they are particularly appropriate for analyzing associations between categorical variables in cross-sectional data [54]. Given the aggregate nature of the dataset, analyses were limited to descriptive and unadjusted tests. This prevented conducting multivariate regression analyses or modeling of adjusted risk ratios or odds ratios, which would have required individual-level (row-wise) data. Despite this limitation, chi-square tests effectively captured significant associations and differences within the available aggregated data, thus meeting the research objectives and ensuring robust descriptive conclusions [55].

#### 2.4.3. Ethical Considerations

Given that the dataset used in this study was publicly available, anonymized, and aggregated, no ethical approval or Institutional Review Board (IRB) permission was required. There was no direct interaction with human subjects or use of personally identifiable data. The publicly disseminated data from GAStat ensured compliance with ethical standards for secondary analyses of de-identified population data. Furthermore, the study aimed to provide essential prevalence estimates and insights to inform evidence-based policy development and to enhance targeted public health interventions for people with mobility difficulties. Therefore, the ethical risk associated with the analysis was minimal, and the research adhered to international standards for ethical secondary data use [56].

## 3. Results

This results section provides a detailed analysis of mobility-related physical difficulties among the Saudi population based on data from the 2017 National Disability Survey. The survey enumerated 20,408,362 Saudi citizens, not including 12,143,974 residents, with a slightly higher number of males (10,396,914) than females (10,011,448). Among this population, a total of 1,445,723 individuals (7.1%) reported having at least one form of functional difficulty. The majority of Saudis (18,962,639 individuals; males = 9,641,679, females = 9,320,960) reported no functional difficulty. Specifically, mobility difficulty—defined as trouble walking or climbing stairs—was prevalent among 833,136 individuals. Of these, 305,867 individuals (36.7%) had mobility difficulty only, while the remaining 527,269 (63.3%) reported mobility difficulty combined with one or more additional functional limitations. Sex-specific analyses revealed that women (459,506) slightly outnumbered men (373,630) in the overall mobility-difficulty group. Moreover, women were notably more represented among individuals with mobility-only difficulty (177,462 women versus 128,405 men) and those with multiple functional difficulties (282,044 women versus 245,225 men). The following tables and figures comprehensively describe the characteristics of this population, including regional differences, severity levels, sociodemographic and familial contexts, causes and duration of difficulty, and utilization patterns for aids and supportive services. Appendix A provide age-group distributions (counts and row percentages) for Saudis reporting a single functional difficulty and for those reporting mobility difficulty plus at least one additional limitation, respectively, thereby enabling life-course comparisons of single- versus multiple-difficulty burdens.

Panel A (Figure 1) shows that the prevalence of difficulty walking or climbing stairs ranges from ≈ 9% in Aseer—the highest of all thirteen administrative areas—to ≈ 3% in Najran, with a steady gradient across intermediate regions. Panel B (Figure 1) indicates that mobility limitations are predominantly mild at the national level; moderate cases form a sizeable secondary tier, while severe limitations represent the smallest stratum—roughly one-fifth of mild cases—underscoring an inverted-pyramid severity structure. In Panel C (Figure 1), disease-related conditions account for well over half of all reported causes, dwarfing every other category; “other” causes and accidents follow distantly, whereas congenital and perinatal factors together constitute a much smaller share. The steep drop-off from disease to the next cause emphasizes the paramount role of chronic illness in Saudi mobility disability. Panel D reveals that crutches are the most frequently used mobility aid, followed by informal help from another person and wheelchairs. Standing devices, “other” aids, and artificial limbs represent only a small fraction of total aid uptake, pointing to a continued reliance on low-technology or personal assistance solutions.

To guide interpretation, we raised 5 questions—each research question is addressed in a specific display. National prevalence and the sex-specific severity gradient (RQ 1) are presented in Table 1 and visualized in Figure 1B. Regional variation in prevalence (RQ 2) appears in the regional rows of Table 1 and is mapped graphically in Figure 1A. Sociodemographic associations with education and marital status (RQ 3) are detailed in Table 2 and Table 3, respectively. Familial context, etiology, and chronicity among those with multiple difficulties (RQ 4) are reported in Table 4A–C. Finally, utilization of mobility aids and Ministry services (RQ 5) is summarized in Table 5 and illustrated in Figure 1D.

The 2017 National Disability Survey enumerated ≈ 20.4 million Saudis, of whom a little more than 1.4 million—roughly one person in fourteen—reported at least one functional difficulty (Table 1, Total row). Within this group, difficulties related specifically to walking or climbing stairs were the most common: almost three-fifths of all individuals with any disability experienced a mobility limitation, most frequently of mild intensity, with progressively smaller shares classified as moderate or severe.

Regional patterns were uneven. The prevalence of any functional difficulty ranged from just under 3 percent in Najran to almost 10 percent in Aseer, underscoring sizeable geographic differences in disability burden. Corresponding mobility-difficulty proportions mirrored this gradient. Sex-specific columns show a consistent but modest male excess nationally (7.3 percent of males vs. 6.9 percent of females); the same pattern held in most regions, although Northern Borders and a few other areas exhibited the opposite or minimal gap. Taken together, the table highlights three actionable findings: (a) a sizeable national pool of persons with mobility limitations that concentrates in certain regions, (b) clear regional outliers that merit targeted investigation and resource allocation, and (c) small but systematic sex differentials that should inform program design.

Table 2 shows that more than one-third of all respondents with mobility difficulties had never attended school, while fewer than one in ten held a university qualification. Illiteracy accounted for over half of affected females but only one-eighth of affected males, a sex gap confirmed by a highly significant χ^2^ test (*p* < 0.001). Conversely, men made up the majority of university-educated cases.

Table 3 indicates that married adults represented almost three-quarters of all mobility-difficulty cases aged ≥15 years. Nonetheless, never-married individuals were proportionally more common among males, whereas widowed status was far more frequent among females; the overall marital-status distribution differed sharply by sex (*p* < 0.001).

Parental relationship (Panel A). Just over half of respondents with both mobility difficulty and at least one additional limitation reported *no parental relationship* to their main caregiver (50.2%). Of those who cited family ties, the most common link was through both parents (16.4%), followed by other relatives (15.8%) and father-side first-degree relatives (11.9%). Men were more likely than women to name father-side relatives (12.7% vs. 11.3%), whereas women more often reported “other relatives” (18.4% vs. 12.2%). These differences were significant, χ^2^(4) = 2489.4, *p* < 0.001. Cause of difficulty (Panel B). Disease was the leading reported cause, accounting for 58.8% of multiple-difficulty cases; it was especially common among women (62.8% vs. 53.1% of men). Congenital conditions ranked a distant second (10.2% overall) and showed little sex disparity. Accident-related causes, though relatively rare overall (traffic 5.8%; other accidents 5.9%), were heavily male-skewed: traffic accidents were cited by 12.2% of men but only 1.1% of women. The overall pattern differed sharply by sex, χ^2^(6) = 19,165.1, *p* < 0.001. Duration (Panel C). Multiple disabilities associated with mobility problems proved overwhelmingly chronic: 91.8% of all cases—and nearly 94% of female cases—had lasted 25 years or longer. Recent-onset limitations (0–4 years) were uncommon (1.4% overall) but twice as prevalent among men as women (2.3% vs. 0.8%). Sex differences across duration bands remained significant, χ^2^(5) = 2990.5, *p* < 0.001.

All in all, most Saudis with co-occurring mobility and other difficulties have extremely long-standing, disease-related conditions and often lack a close parental caregiver. Pronounced sex differences—men’s greater exposure to accident-related causes and women’s higher concentration in the longest-duration category—underscore the need for gender-sensitive, lifelong support services.

Among Saudi residents who reported both mobility difficulties and at least one additional functional limitation, nearly half (46.8%) indicated no parental relationship to their primary caregiver, whereas smaller proportions cited first-degree relatives on the father’s side (13.6%), on the mother’s side (3.8%), both parents (17.4%), or other relatives (18.5%). In sex-specific terms, men were somewhat more likely than women to report “other relatives” (20.2% vs. 17.1%), while women were slightly more likely to name first-degree father’s-side relatives (14.4% vs. 12.6%). The overall distribution differed significantly by sex, χ^2^(4) = 1056.3, *p* < 0.001. With respect to cause, disease dominated, accounting for 56.3% of multiple-difficulty cases; it was especially common among women (61.5% vs. 50.2% of men). Congenital conditions formed the next largest group (12.6% overall), again with a higher share among men (15.6% vs. 9.9%). Traffic accidents were rare overall (3.4%) but markedly concentrated among men (6.5% vs. 0.7% of women). These sex disparities produced a highly significant association, χ^2^(6) = 20,174.0, *p* < 0.001. Regarding duration, the burden was overwhelmingly chronic: 90.1% of respondents had lived with their mobility difficulties for 25 years or longer. Short-duration problems (0–9 years combined) represented fewer than 4% of cases. Men were modestly overrepresented in the 5–9-year (2.4% vs. 1.7%) and 20–24-year (3.9% vs. 1.6%) bands, whereas women predominated in the longest-duration category (91.4% vs. 88.5%). The sex distribution across duration categories was again significant, χ^2^(5) = 3232.9, *p* < 0.001.

Taken together, Table 5 shows that multiple disabilities linked to mobility problems in Saudi Arabia are usually long-standing, most often disease-related, and display distinct sex patterns in cause and familial context—observations that have clear implications for chronic-care planning and gender-sensitive support services.

Mobility aids (Table 6, Panel A). More than one-quarter of all respondents relied on help from another person (25.3%), while nearly half used crutches (47.7%) and almost one-quarter used a wheelchair (22.6%). Sex differences were pronounced (χ^2^(5) = 3267.0, *p* < 0.001). Men showed greater reliance on crutches (51.8% of male aid users vs. 45.1% of females) and wheelchairs (21.7% vs. 23.1%), whereas women more often reported personal assistance (27.1% vs. 22.4%). Use of artificial limbs, standing devices, and “other” aids each accounted for ≤2% of cases in both sexes.

Ministry services (Table 6, Panel B). Fully 83.8% of difficulty-reporting respondents received a physical (medical rehabilitation) service, and 9.3% received in-kind benefits. Less than four percent accessed accommodation, daycare, or home care programs combined. The service mix differed by sex (χ^2^(5) = 2842.0, *p* < 0.001): women were more likely than men to obtain physical services (86.2% vs. 81.0%), whereas men more often reported in-kind benefits (11.4% vs. 7.6%) and home care (1.9% vs. 2.3% for women, but higher in raw numbers among men). “Other” services represented a small share overall (3.4%).

Most Saudis with mobility limitations rely on a combination of personal aid (human assistance) and standard mobility devices, with sizeable sex disparities in specific aids. Ministry service uptake is dominated by physical rehabilitation support, suggesting that auxiliary programs such as accommodation and daycare remain underutilized. These findings highlight where resource expansion—particularly in home- and community-based services—could have the greatest impact.

## 4. Discussion

The purpose of this investigation was to generate the first population-based portrait of mobility difficulty in Saudi Arabia, deploying the 2017 National Disability Survey to answer five research questions that spanned prevalence, geography, socio-demographic correlates, contextual characteristics, and service use. The analyses revealed five key findings. First, 7.1% of Saudi citizens reported at least one mobility limitation; mild cases predominated, and prevalence was slightly higher in men (7.3%) than women (6.9%) (i.e., with at least one reported difficulty: 7.08%; with mobility difficulty and/or other difficulties: 4.08%; and with mobility difficulty alone: 1.50%). Second, prevalence varied markedly across the thirteen regions, ranging from a high of 9.4% in Aseer to a low of 2.9% in Najran. Third, mobility difficulty clustered among adults with low education and those who were widowed or never married, with sharper sex gaps at both extremes of attainment and marital status. Fourth, among respondents who had mobility difficulties plus at least one additional limitation, chronic disease was the leading cause, most cases had persisted for ≥25 years, and women outnumbered men in the longest-duration band. Fifth, almost half of all affected Saudis relied on crutches, one quarter on personal assistance, and four-fifths received physical rehabilitation services, yet more sophisticated aids and community programs were sparsely used.

The prevalence estimates reported here position Saudi Arabia within the mid-range of international figures. For comparison, in the United States, mobility limitation is evident in 11% of adults aged 50–64 [10], while approximately 18% of Estonia’s working-age population (20–79 years) reports significant activity restrictions [11]. Although the Saudi prevalence (7.1%) across the broader adult population appears lower, it nonetheless signifies substantial unmet needs, particularly given the extremely low physical activity levels prevalent in Saudi society [37]. Interestingly, the slight excess of male cases contrasts markedly with data from Western countries, where women consistently report higher rates of mobility impairment [14]. This difference could reflect distinct socio-cultural and occupational patterns in Saudi Arabia, particularly men’s greater exposure to physically demanding jobs and higher incidence of severe traffic-related injuries, which counterbalance the biological resilience typically observed among women. These patterns underline the importance of context-specific analyses in understanding epidemiological trends and formulating targeted interventions.

Regional disparities in mobility limitations observed in the current study are notably pronounced, with substantial variations between regions such as Aseer (9.4%) and Najran (2.9%). These variations likely reflect a complex interplay between demographic composition, environmental conditions, and cultural practices. For instance, previous qualitative research conducted in cities such as Jeddah and Riyadh identifies extreme climate conditions, fragmented pedestrian networks, and restrictive social norms—particularly for women—as major barriers to active mobility [38,39]. Aseer’s notably higher prevalence may reflect its mountainous geography, challenging accessibility, and possibly a higher proportion of older adults. Conversely, Najran’s lower burden may be attributable to younger demographic profiles, denser urban planning, and possibly different cultural attitudes toward outdoor mobility. These findings underscore the necessity of developing region-specific, mobility-friendly urban designs and culturally sensitive policies that enhance accessibility and promote active lifestyles across different Saudi regions.

Sociodemographic correlates of mobility difficulty, specifically lower educational attainment and widowhood, emerged prominently in this study, aligning closely with international research findings. Similar associations have been documented in Estonia and among racially and ethnically stratified cohorts in the United States, where lower socioeconomic status and weaker social support structures consistently amplify mobility limitations [11,14]. These results resonate with social-ecological models that interpret mobility as an outcome shaped by the intersection of individual capabilities, community resources, and broader structural conditions [33]. In the Saudi context, lower educational attainment may limit health literacy, restrict access to preventive healthcare services, and reduce individuals’ capacity to adapt effectively to mobility impairments within a predominantly vehicle-dependent society. Likewise, widowhood and unmarried status may reflect diminished social networks and reduced access to caregiving resources, exacerbating existing vulnerabilities. Such findings call for multi-sectoral approaches that integrate social services, educational initiatives, and community-based support structures to address mobility-related inequities comprehensively.

The dominance of chronic disease as a primary cause of mobility difficulties, often persisting for decades, aligns strongly with global evidence linking obesity, multimorbidity, and sedentary lifestyles to long-term functional limitations [5]. The gender disparity noted in the longest-duration cases, with women disproportionately represented, also echoes international literature highlighting longer disability durations and slower recovery rates among women [4]. This extended duration of disability among Saudi women may also reflect socio-cultural contexts in which women’s healthcare needs, especially chronic care and rehabilitation, might be inadequately addressed or delayed due to access barriers. Importantly, research from Saudi trauma cohorts underscores that family cohesion significantly moderates rehabilitation outcomes, implying that supportive familial environments can facilitate recovery and reduce prolonged disability [46]. Collectively, these insights highlight the need for gender-sensitive and family-centric rehabilitation models, emphasizing community-based preventive strategies, comprehensive chronic disease management, and increased access to tailored rehabilitative services to mitigate prolonged mobility limitations.

National Global Burden of Disease data identify musculoskeletal disorders—especially low-back pain and knee osteoarthritis—as the second-largest cause of years-lived-with-disability in Saudi Arabia, surpassed only by cardiovascular disease [57]. Recent regional work confirms the clinical impact: among outpatients with musculoskeletal conditions in Jizan, greater pain severity and multimorbidity sharply increase dependence in activities of daily living, with women overrepresented in the high-dependency group [58]. A systematic review further singles out knee osteoarthritis as the most prevalent joint disorder limiting ambulation nationwide [59]. Parallel epidemiology shows type 2 diabetes—affecting roughly one quarter of Saudi adults and disproportionately women of working age—compromises gait via neuropathy, obesity, and vascular disease [60]. First, a coordinated noncommunicable disease program should integrate weight control, glycemic optimization, and joint-friendly resistance plus aerobic exercise delivered through primary care hubs and female-only, climate-controlled fitness centres. Similar multimodal regimens reduce pain and slow functional decline by 20–30% in comparable cohorts [4]. Second, early knee-OA screening (e.g., targeted X-ray for women ≥ 45 y with BMI > 30 kg m^−2^) coupled with digital self-management platforms that cue home-based strength and balance routines can raise six-minute-walk distance by >10% within 12 weeks; Gulf trials using smart wheelchairs report comparable functional gains when assistive technology is paired with coaching [43]. Embedding these actions in Vision 2030’s preventive-health pillar would measurably curb the chronic-disease toll that the present survey attributes predominantly to women.

The utilization patterns for mobility aids and rehabilitation services identified in this study illuminate critical gaps in assistive-technology infrastructure. Despite emerging evidence supporting advanced mobility technologies such as smart wheelchairs in Saudi contexts [43] and calls for structured frameworks to enhance assistive-technology provision [44], current data indicate heavy reliance on basic aids such as crutches and direct human assistance. These traditional solutions, although effective for mild or temporary limitations, provide limited autonomy and independence for those with severe and persistent mobility impairments. Moreover, the high dependency on Ministry-provided physical rehabilitation services contrasts sharply with the limited engagement in home care programs, community-based rehabilitation, and daycare centres. Given that functional mobility and balance confidence have been demonstrated as critical predictors of long-term disability among Saudi older adults [42], the current reliance on simplistic assistive devices and centralized physical rehabilitation appears insufficient. Therefore, the development and promotion of more sophisticated, accessible, and community-integrated rehabilitation technologies and programs are therefore crucial. Policymakers and healthcare providers must prioritize broadening the scope and reach of rehabilitative services, integrating advanced assistive technologies, and enhancing community-based supports to improve overall functional outcomes and independence among Saudis with mobility difficulties.

### 4.1. Strengths and Limitations

This study has several strengths. For instance, this design is particularly suited for epidemiological investigations aiming to determine the distribution, patterns, and associated factors of specific health conditions within a clearly defined population at one point in time [61]. A cross-sectional design was chosen due to its practicality, efficiency, and ability to yield reliable prevalence estimates necessary for healthcare planning, policy formulation, and resource allocation [62,63]. This approach allowed for a comprehensive snapshot of the magnitude and distribution of mobility difficulties across multiple demographic and geographic subgroups, facilitating targeted interventions and informed decision-making [55]. However, as with all cross-sectional designs, limitations exist, such as the inability to establish causal relationships or temporality between risk factors and outcomes [64]. Nevertheless, the descriptive value of this methodology outweighs these constraints, as the primary objective of the study was not to infer causation but rather to precisely describe the prevalence, severity, and demographic distribution of mobility-related disabilities. Additionally, the large-scale, population-based nature of this national disability survey significantly enhanced the generalizability and relevance of the findings to the Saudi population, ensuring robust and contextually valid conclusions [55,61]. More importantly, a key restriction of the public dataset is that age was tabulated only for the binary categories ‘with any difficulty’ versus ‘without difficulty’; because age was not cross-classified by specific difficulty type, we were unable to calculate age-specific prevalence of mobility difficulty per se or identify which age strata contribute most to the national burden—an analytic gap that future studies using individual-level microdata will need to address.

Furthermore, the current study is subject to several methodological limitations that should be considered when interpreting the findings. Firstly, the analysis relied exclusively on cross-sectional data from the 2017 National Disability Survey, restricting our capacity to draw causal inferences or track longitudinal patterns of mobility difficulties [4]. This design limitation also precludes examination of temporal changes in prevalence and service utilization, both of which may have evolved significantly since data collection, particularly in response to recent public health and infrastructure initiatives [38]. Furthermore, although the dataset was nationally representative, it lacked detailed clinical information regarding underlying chronic conditions, severity gradations, and precise functional measures such as walking speed or balance scores [10]. Such granularity would have allowed more precise characterization of mobility impairments and their practical implications. Finally, due to data limitations, this analysis could not address psychological and cognitive dimensions of mobility difficulties, despite evidence suggesting their substantial impact on functional limitations [6]. These gaps underscore the need for richer, longitudinal, and multi-dimensional datasets to advance understanding of mobility difficulties within Saudi populations.

### 4.2. Future Directions

Future research should prioritize longitudinal cohort studies and mixed-method designs to capture trajectories of mobility difficulty over time, enabling causal analysis of risk factors and outcomes. Prospective studies with repeated assessments would clarify temporal relationships between chronic diseases, lifestyle factors, and the onset or progression of mobility limitations [5]. Qualitative and mixed-method research could also deepen understanding of cultural and environmental factors unique to Saudi Arabia, especially concerning how social norms, gender roles, and urban design jointly shape active-mobility practices [38,39]. Expanding research to explore psychological and cognitive correlates of mobility difficulty would further illuminate intervention targets, particularly for older adults experiencing concurrent cognitive decline [6]. Finally, evaluation studies of advanced assistive technologies and community-based rehabilitation programs could identify effective strategies to enhance autonomy and reduce dependence on simplistic mobility aids, particularly for individuals with severe and persistent difficulties [43].

### 4.3. Implications

The findings from this investigation have several critical implications for public health policy, clinical practice, and community planning in Saudi Arabia. The demonstrated regional and sociodemographic disparities imply the urgent need for targeted, geographically and culturally sensitive interventions. Public health initiatives should prioritize regions with high mobility-difficulty prevalence and focus on infrastructure improvements, such as pedestrian-friendly environments and climate-adapted designs, to promote active living [38]. Healthcare systems should enhance preventive and rehabilitative services, particularly targeting chronic diseases and providing gender-sensitive and family-centered care that acknowledges cultural contexts influencing health behaviors [46]. Furthermore, investments in assistive-technology infrastructure are essential, moving beyond basic aids toward advanced devices that significantly improve quality of life and autonomy [44]. Integrating these strategies into Saudi Arabia’s broader public-health and development frameworks, aligned with Vision 2030 objectives, will be critical for addressing the complex needs of individuals with mobility impairments. Moreover, our findings provide essential context for planning rehabilitation services, accessibility initiatives, and region-specific preventive strategies aimed at reducing the onset and progression of mobility-related disability in Saudi Arabia. Lastly, they suggest that educational attainment and marital circumstances are strongly intertwined with the sex profile of mobility-related disability in Saudi Arabia, pointing to distinct life-course and social-support considerations for men and women.

### 4.4. Conclusions

This study provided the first detailed epidemiological analysis of mobility-related physical difficulties among Saudi citizens, highlighting significant prevalence, notable regional disparities, and pronounced sociodemographic variations. Chronic diseases emerged as the leading cause, particularly among women, emphasizing the long-term burden of mobility impairment. The observed reliance on basic mobility aids and limited use of advanced rehabilitation services underscore the need for substantial investments in technology and infrastructure. Addressing these gaps through targeted interventions, culturally sensitive health programs, and strengthened rehabilitative frameworks will significantly enhance mobility, independence, and quality of life for Saudi citizens experiencing mobility difficulties.

## Figures and Tables

**Figure 1 healthcare-13-01804-f001:**
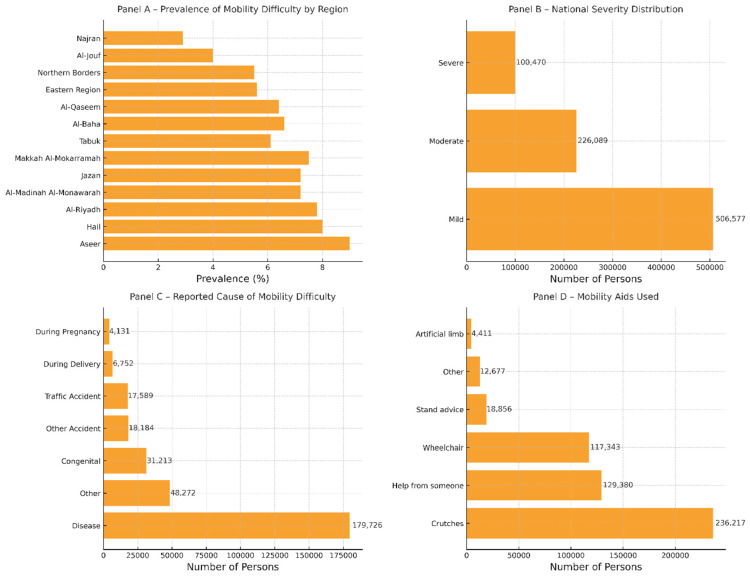
Key epidemiologic and care characteristics of mobility-related physical difficulty in Saudi Arabia, 2017 National Disability Survey.

**Table 1 healthcare-13-01804-t001:** Prevalence of mobility-related physical difficulty by region and sex—Saudi Arabia, 2017.

Area	Total Pop	Any Difficulty (*n*, %)	Male Pop	Male Diff (*n*, %)	Female Pop	Female Diff (*n*, %)	Mild (N, %)	Moderate (*n*, %)	Severe (*n*, %)
Al-Riyadh	4,658,322	363,351 **(7.8%)**	2,414,876	196,158 **(8.1%)**	2,243,446	167,193 **(7.5%)**	116,118 **(32%)**	46,367 **(13%)**	24,594 **(7%)**
Makkah	4,516,577	336,705 **(7.5%)**	2,290,489	175,925 **(7.7%)**	2,226,088	160,780 **(7.2%)**	136,801 **(41%)**	56,405 **(17%)**	24,891 **(7%)**
Al-Madinah	1,376,244	98,905 **(7.2%)**	690,678	52,771 **(7.6%)**	685,566	46,134 **(6.7%)**	31,151 **(31%)**	17,033 **(17%)**	6987 **(7%)**
Al-Qaseem	1,009,543	65,021 **(6.4%)**	511,037	33,722 **(6.6%)**	498,506	31,299 **(6.3%)**	21,463 **(33%)**	8848 **(14%)**	5826 **(9%)**
Eastern Region	3,140,362	177,064 **(5.6%)**	1,634,246	88,518 **(5.4%)**	1,506,116	88,546 **(5.9%)**	56,274 **(32%)**	29,320 **(17%)**	12,376 **(7%)**
Aseer	1,750,131	164,441 **(9.4%)**	864,951	83,170 **(9.6%)**	885,180	81,271 **(9.2%)**	60,661 **(37%)**	25,821 **(16%)**	9796 **(6%)**
Tabouk	722,664	43,943 **(6.1%)**	371,087	23,592 **(6.4%)**	351,577	20,351 **(5.8%)**	15,579 **(35%)**	6520 **(15%)**	1690 **(4%)**
Hail	538,099	43,438 **(8.1%)**	266,259	21,250 **(8.0%)**	271,840	22,188 **(8.2%)**	14,738 **(34%)**	9002 **(21%)**	3948 **(9%)**
Northern Borders	288,921	12,893 **(4.5%)**	145,415	6246 **(4.3%)**	143,506	6647 **(4.6%)**	4801 **(37%)**	2581 **(20%)**	1666 **(13%)**
Jazan	1,207,269	87,348 **(7.2%)**	610,691	46,257 **(7.6%)**	596,578	41,091 **(6.9%)**	28,169 **(32%)**	14,383 **(16%)**	5416 **(6%)**
Najran	438,041	12,618 **(2.9%)**	220,335	6835 **(3.1%)**	217,706	5783 **(2.7%)**	7658 **(61%)**	1056 **(8%)**	627 **(5%)**
Al-Baha	382,438	25,316 **(6.6%)**	184,080	13,084 **(7.1%)**	198,358	12,232 **(6.2%)**	9257 **(37%)**	4248 **(17%)**	1760 **(7%)**
Al-Jouf	379,751	14,680 **(3.9%)**	192,770	7707 **(4.0%)**	186,981	6973 **(3.7%)**	3907 **(27%)**	4505 **(31%)**	893 **(6%)**
Total	20,408,362	1,445,723 **(7.1%)**	10,396,914	755,235 **(7.3%)**	10,011,448	690,488 **(6.9%)**	506,577 **(35%)**	226,089 **(16%)**	100,470 **(7%)**

Note. Numbers represent population counts; percentages in bold parentheses are prevalence rates. For ‘Any Difficulty,’ prevalence is calculated as the number of individuals with physical difficulty divided by the total population in each region. For ‘Male Diff’ and ‘Female Diff,’ prevalence is the proportion of affected within the respective sex. Severity percentages are the proportion of each severity category among all affected in that region. Percentages are rounded to one decimal place. Data sources: Baseline Data 3 and Indicator 1 (2017 National Disability Survey).

**Table 2 healthcare-13-01804-t002:** Prevalence of mobility-related physical difficulty by educational status (≥10 years)—Saudi Arabia, 2017.

Category	Total *n* (%)	Male *n* (%)	Female *n* (%)
Illiterate	107,085 (36.1%)	15,442 (12.5%)	91,643 (52.9%)
Read and Write	42,204 (14.2%)	15,796 (12.8%)	26,408 (15.2%)
Primary	40,272 (13.6%)	21,849 (17.7%)	18,423 (10.6%)
Intermediate	26,502 (8.9%)	13,342 (10.8%)	13,160 (7.6%)
Secondary/Equivalent	40,776 (13.8%)	26,139 (21.2%)	14,637 (8.5%)
Pre-Univ. Diploma	11,414 (3.9%)	9533 (7.7%)	1881 (1.1%)
University and Higher	28,194 (9.5%)	21,151 (17.2%)	7043 (4.1%)

χ^2^(6) = 66,081, *p* < 0.001. Note. Entries are counts of individuals reporting difficulty walking or climbing stairs, with row percentages in parentheses. Row percentages are calculated within each sex and within the total mobility-difficulty population for the panel. χ^2^ tests compare the distribution of categories between males and females. Data source: Indicators 4 and 5 (2017 National Disability Survey). Percentages are rounded to one decimal place.

**Table 3 healthcare-13-01804-t003:** Prevalence of mobility-related physical difficulty by marital status (≥15 years)—Saudi Arabia, 2017.

Category	Total *n* (%)	Male *n* (%)	Female *n* (%)
Never Married	21,346 (7.3%)	12,568 (10.4%)	8778 (5.2%)
Married	207,147 (71.2%)	103,922 (86.2%)	103,225 (60.6%)
Divorced	8536 (2.9%)	2056 (1.7%)	6480 (3.8%)
Widowed	53,817 (18.5%)	2058 (1.7%)	51,759 (30.4%)

χ^2^(3) = 41,608, *p* < 0.001. Note. Entries are counts of individuals reporting difficulty walking or climbing stairs, with row percentages in parentheses. Row percentages are calculated within each sex and within the total mobility-difficulty population for the panel. χ^2^ tests compare the distribution of categories between males and females. Data source: Indicators 4 and 5 (2017 National Disability Survey). Percentages are rounded to one decimal place.

**Table 4 healthcare-13-01804-t004:** Context of mobility difficulty: parental relationship, cause, and duration—individuals reporting walking/climbing difficulty, Saudi Arabia 2017. Panel (**A**) parental relationship. Panel (**B**) reported cause of difficulty. Panel (**C**) duration of difficulty (years).

(A)			
**Category**	**Total (*n* = 305,867)**	**Male (*n* = 128,405)**	**Female (*n* = 177,462)**
First-degree relatives on father’s side	36,480 (11.9%)	16,370 (12.7%)	20,110 (11.3%)
First-degree relatives on mother’s side	17,381 (5.7%)	8041 (6.3%)	9340 (5.3%)
First-degree relatives Father and mother hand	50,028 (16.4%)	23,251 (18.1%)	26,777 (15.1%)
Other relatives	48,397 (15.8%)	15,699 (12.2%)	32,698 (18.4%)
Not related	153,581 (50.2%)	65,044 (50.7%)	88,537 (49.9%)
χ^2^(4) = 2489.4, *p* < 0.001.			
(B)			
**Category**	**Total (*n* = 305,867)**	**Male (*n* = 128,405)**	**Female (*n* = 177,462)**
Congenital	31,213 (10.2%)	12,854 (10.0%)	18,359 (10.3%)
During Pregnancy	4131 (1.4%)	708 (0.6%)	3423 (1.9%)
During Delivery	6752 (2.2%)	2279 (1.8%)	4473 (2.5%)
Traffic Accident	17,589 (5.8%)	15,618 (12.2%)	1971 (1.1%)
Other Accident	18,184 (5.9%)	9522 (7.4%)	8662 (4.9%)
Disease	179,726 (58.8%)	68,206 (53.1%)	111,520 (62.8%)
Other	48,272 (15.8%)	19,218 (15.0%)	29,054 (16.4%)
χ^2^(6) = 19,165.1, *p* < 0.001.			
(C)			
**Category**	**Total (*n* = 305,867)**	**Male (*n* = 128,405)**	**Female (*n* = 177,462)**
0–4	4311 (1.4%)	2946 (2.3%)	1365 (0.8%)
5–9	5109 (1.7%)	2207 (1.7%)	2902 (1.6%)
10–14	5601 (1.8%)	2648 (2.1%)	2953 (1.7%)
15–19	5115 (1.7%)	3344 (2.6%)	1771 (1.0%)
20–24	4912 (1.6%)	2735 (2.1%)	2177 (1.2%)
25+	280,819 (91.8%)	114,525 (89.2%)	166,294 (93.7%)

χ^2^(5) = 2990.5, *p* < 0.001. Note. Counts are the number of respondents who reported walking/climbing difficulty. Percentages (bold, in parentheses) are column percentages within total, male, or female. χ^2^ tests compare male–female distributions.

**Table 5 healthcare-13-01804-t005:** Multiple-difficulty characteristics among Saudi residents with mobility-related physical difficulty, by parental relationship, cause, and duration (2017 National Disability Survey). Panel (**A**)—parental relationship. Panel (**B**)—cause of difficulty. Panel (**C**)—duration of difficulty.

(A)			
**Category**	**Total *n* (%)**	**Male *n* (%)**	**Female *n* (%)**
First-degree relatives on father’s side	71,480 (13.6%)	30,952 (12.6%)	40,528 (14.4%)
First-degree relatives on mother’s side	19,919 (3.8%)	8590 (3.5%)	11,329 (4.0%)
First-degree relatives Father and mother hand	91,512 (17.4%)	42,421 (17.3%)	49,091 (17.4%)
Other relatives	97,751 (18.5%)	49,455 (20.2%)	48,296 (17.1%)
Not related	246,607 (46.8%)	113,807 (46.4%)	132,800 (47.1%)
(B)			
**Category**	**Total *n* (%)**	**Male *n* (%)**	**Female *n* (%)**
Congenital	66,237 (12.6%)	38,241 (15.6%)	27,996 (9.9%)
During Pregnancy	7041 (1.3%)	4079 (1.7%)	2962 (1.1%)
During Delivery	19,727 (3.7%)	10,027 (4.1%)	9700 (3.4%)
Traffic Accident	17,956 (3.4%)	15,995 (6.5%)	1961 (0.7%)
Other Accident	31,664 (6.0%)	15,040 (6.1%)	16,624 (5.9%)
Disease	296,740 (56.3%)	123,154 (50.2%)	173,586 (61.5%)
Other	87,904 (16.7%)	38,689 (15.8%)	49,215 (17.4%)
(C)			
**Category**	**Total *n* (%)**	**Male *n* (%)**	**Female *n* (%)**
0–4	9333 (1.8%)	3962 (1.6%)	5371 (1.9%)
5–9	10,857 (2.1%)	5980 (2.4%)	4877 (1.7%)
10–14	9994 (1.9%)	4808 (2.0%)	5186 (1.8%)
15–19	8118 (1.5%)	3805 (1.6%)	4313 (1.5%)
20–24	13,928 (2.6%)	9546 (3.9%)	4382 (1.6%)
25+	475,039 (90.1%)	217,124 (88.5%)	257,915 (91.4%)

Note. Percentages are row percentages within Total, Male, or Female difficulty cases. Sex differences tested by χ^2^: parental relationship χ^2^(4) = 1056.3, cause χ^2^(6) = 20,174.0, duration χ^2^(5) = 3232.9; all *p* < 0.001. Data: Indicators 10–12, 2017 National Disability Survey.

**Table 6 healthcare-13-01804-t006:** Use of mobility aids and ministry services among Saudi residents reporting walking/climbing difficulty, 2017 National Disability Survey. Panel (**A**)—mobility aids. Panel (**B**)—ministry services.

(A)							
**Sex**	**Total**	**Other**	**Help Someone**	**Artificial Limbs**	**Stand** **Device**	**Wheelchair**	**Crutches**
Male	194,866	3297 (1.7%)	43,641 (22.4%)	1381 (0.7%)	3252 (1.7%)	42,285 (21.7%)	101,010 (51.8%)
Female	304,989	6574 (2.2%)	82,697 (27.1%)	711 (0.2%)	6965 (2.3%)	70,442 (23.1%)	137,600 (45.1%)
Total	499,855	9871 (2.0%)	126,338 (25.3%)	2092 (0.4%)	10,217 (2.0%)	112,727 (22.6%)	238,610 (47.7%)
(B)							
**Sex**	**Total**	**Other**	**Accommodation**	**Day Care**	**Home Care**	**Physical**	**In-Kind**
Male	161,007	7250 (4.5%)	498 (0.3%)	1354 (0.8%)	3073 (1.9%)	130,413 (81.0%)	18,419 (11.4%)
Female	192,607	4829 (2.5%)	547 (0.3%)	2188 (1.1%)	4397 (2.3%)	166,046 (86.2%)	14,600 (7.6%)
Total	353,614	12,079 (3.4%)	1045 (0.3%)	3542 (1.0%)	7470 (2.1%)	296,459 (83.8%)	33,019 (9.3%)

Note. Counts represent survey respondents with walking/climbing difficulties who reported using each aid or receiving each service. Percentages in parentheses are within-row percentages (total, male, female). Sex differences in distribution were significant for both panels: Aids χ^2^(5) = 3267.0, *p* < 0.001; Services χ^2^(5) = 2842.0, *p* < 0.001.

## Data Availability

The data that support the findings of this study are available in the General Authority for Statistics, Saudi Arabia, at https://www.stats.gov.sa/en/home, Accessed on 1 May 2025. These data were derived from the following resources available in the public domain:—Social Statistics, https://www.stats.gov.sa/en/statistics?index=119025, Accessed on 1 May 2025.

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
