# Peer review of "The Epidemiology of Mobility Difficulty in Saudi Arabia: National Estimates, Severity Levels, and Sociodemographic Differentials"

_healthcare, 2025, doi:10.3390/healthcare13151804_

Round 1

Reviewer 1 Report

Comments and Suggestions for Authors

Thank you for the opportunity to review this article. Please find my comments.

 INTRODUCTION

The text in the 'Introduction' section is very long, which makes reading tedious. Please rewrite this section.

METHODS

-Lines 239-247: “This design is particularly suited for epidemiological investigations aiming to determine the distribution, patterns, and associated factors of specific health conditions within a clearly defined population at one point in time (Levin, 2019). A cross-sectional design was chosen due to its practicality, efficiency, and ability to yield reliable prevalence estimates necessary for healthcare planning, policy formulation, and resource allocation (Setia, 2016; Wang & Cheng, 2020). This approach allowed for a comprehensive snapshot of the magnitude and distribution of mobility difficulties across multiple demo-graphic and geographic subgroups, facilitating targeted interventions and informed decision-making (Sedgwick, 2014).”  This information could be moved to “Discussion” section, highlighting it as one of the strengths of the study. Please think about this.

-Lines 248-256: “However, as with all cross-sectional designs, limitations exist, such as the inability to establish causal relationships or temporality between risk factors and outcomes (Carlson & Morrison, 2023). Nevertheless, the descriptive value of this methodology outweighs these constraints, as the primary objective of the study was not to infer causation but rather to precisely describe the prevalence, severity, and demographic distribution of mobility-related disabilities. Additionally, the large-scale, population-based nature of this national disability survey significantly enhanced the generalizability and relevance of the findings to the Saudi population, ensuring robust and contextually valid conclusions (Levin, 2019; Sedgwick, 2014).” This information could be moved to “Discussion” section, highlighting this as one weakness of the study of the study. Please think about this.

-Lines 258-259: “The sample for this study was drawn from the Disability Survey 2017 conducted by the General Authority for Statistics (GAStat), Saudi Arabia.” /Lines 284-285: “This study utilized secondary data derived from the 2017 Disability Survey conducted…” / Lines 308-310: “The data utilized for this study were publicly available secondary data obtained from the General Authority for Statistics (GAStat), Kingdom of Saudi Arabia, from the Disability Survey 2017…” Please, exclude repetitive text.

-How many questions are in the questionnaire? Please include information in the text.

-The questionnaire could be available as an Appendix.

RESULTS

-Lines 350-351: “This results section provides a detailed analysis of mobility-related physical difficulties among the Saudi population”  OR  “This results section provides a detailed analysis of mobility-related physical difficulties in Saudi Arabia.” Were only Saudi citizens included in the study? Please, think about this and clarify the information in the text.

-How many young adults, adults and elderly people were included in the study? Please include information about age. It is relevant to answer questions like “Considering Saudi context, was the prevalence of difficulty walking or climbing stairs more common only in older people? Please think about this.

-Lines 420-423: “These figures provide essential context for planning rehabilitation services, accessibility initiatives, and region-specific preventive strategies aimed at reducing the onset and progression of mobility-related disability in Saudi Arabia.” This information could be moved to “Discussion” section.

-Lines 440-442: “These patterns suggest that educational attainment and marital circumstances are strongly intertwined with the sex profile of mobility-related disability in Saudi Arabia, pointing to distinct life-course and social-support considerations for men and women.” This information could be moved to “Discussion” section.

Author Response

Reviewer 1

Thank you for the opportunity to review this article. Please find my comments.

Dear Colleague,

Thank you for your detailed, constructive critique of our manuscript. Your insights were especially valuable. We have addressed each of your points—incorporated in blue font within the revised manuscript. We believe these changes strengthen both the scientific rigour and practical relevance of the work, and we are grateful for your guidance.

Sincerely,

Authors

 INTRODUCTION

The text in the 'Introduction' section is very long, which makes reading tedious. Please rewrite this section.

Thank you, we reduced the introduction from 2200 words to 1300 words.

METHODS

-Lines 239-247: “This design is particularly suited for epidemiological investigations aiming to determine the distribution, patterns, and associated factors of specific health conditions within a clearly defined population at one point in time (Levin, 2019). A cross-sectional design was chosen due to its practicality, efficiency, and ability to yield reliable prevalence estimates necessary for healthcare planning, policy formulation, and resource allocation (Setia, 2016; Wang & Cheng, 2020). This approach allowed for a comprehensive snapshot of the magnitude and distribution of mobility difficulties across multiple demo-graphic and geographic subgroups, facilitating targeted interventions and informed decision-making (Sedgwick, 2014).”  This information could be moved to “Discussion” section, highlighting it as one of the strengths of the study. Please think about this.

Thank you. We moved to the beginning of the limitations sections because we also think it is reasonable to highly the strengths of our study and then move to its limitations.

-Lines 248-256: “However, as with all cross-sectional designs, limitations exist, such as the inability to establish causal relationships or temporality between risk factors and outcomes (Carlson & Morrison, 2023). Nevertheless, the descriptive value of this methodology outweighs these constraints, as the primary objective of the study was not to infer causation but rather to precisely describe the prevalence, severity, and demographic distribution of mobility-related disabilities. Additionally, the large-scale, population-based nature of this national disability survey significantly enhanced the generalizability and relevance of the findings to the Saudi population, ensuring robust and contextually valid conclusions (Levin, 2019; Sedgwick, 2014).” This information could be moved to “Discussion” section, highlighting this as one weakness of the study of the study. Please think about this.

Thank you. Also, moved to the limitations section.

-Lines 258-259: “The sample for this study was drawn from the Disability Survey 2017 conducted by the General Authority for Statistics (GAStat), Saudi Arabia.” /Lines 284-285: “This study utilized secondary data derived from the 2017 Disability Survey conducted…” / Lines 308-310: “The data utilized for this study were publicly available secondary data obtained from the General Authority for Statistics (GAStat), Kingdom of Saudi Arabia, from the Disability Survey 2017…” Please, exclude repetitive text.

Thank you. We checked and remove the repetitive use of General Authority for Statistics and Disability Survey 2017.

-How many questions are in the questionnaire? Please include information in the text.

Thank you, added.

-The questionnaire could be available as an Appendix.

Included as supplementary file and cited as Appendix 1.

RESULTS

-Lines 350-351: “This results section provides a detailed analysis of mobility-related physical difficulties among the Saudi population”  OR  “This results section provides a detailed analysis of mobility-related physical difficulties in Saudi Arabia.” Were only Saudi citizens included in the study? Please, think about this and clarify the information in the text.

Thank you. Yes, I think it is clear in the third line we mentioned excluding residents. They were not included in the detailed disability survey.

-How many young adults, adults and elderly people were included in the study? Please include information about age. It is relevant to answer questions like “Considering Saudi context, was the prevalence of difficulty walking or climbing stairs more common only in older people? Please think about this.

Thank you. Could you please refer to Table 4 Panel C, if you do not mind. The table shows the distribution of population with mobility difficulty by age and sex. Since it is an epidemiological study so we presented this as part of the results other than the method since the survey was inclusive of all ages.

-Lines 420-423: “These figures provide essential context for planning rehabilitation services, accessibility initiatives, and region-specific preventive strategies aimed at reducing the onset and progression of mobility-related disability in Saudi Arabia.” This information could be moved to “Discussion” section.

Thank you. Moved to Implications.

-Lines 440-442: “These patterns suggest that educational attainment and marital circumstances are strongly intertwined with the sex profile of mobility-related disability in Saudi Arabia, pointing to distinct life-course and social-support considerations for men and women.” This information could be moved to “Discussion” section.

Thank you. Also moved to Implications.

Reviewer 2 Report

Comments and Suggestions for Authors

Dear authors,

The manuscript entitled “The Epidemiology of Mobility Difficulty in Saudi Arabia: National Estimates, Severity Levels, and Sociodemographic Differentials” is bringing into attention a major component of the dimension of disability.

While the topic is interesting and the quality of the manuscript is very good, there are some aspects that should be addressed: 

  1. Please pay attention to typo, spaces, and overall aspect of the manuscript.
  2. Regarding the Methods section- Design, I recommend you add the following paragraph at the end, in the discussion section- Limitations: However, as with all cross-sectional designs, limitations exist, such as the inability to establish causal relationships or temporality between risk factors and outcomes (Carlson & Morrison, 2023). Nevertheless, the descriptive value of this methodology outweighs these constraints, as the primary objective of the study was not to infer causation but rather to precisely describe the prevalence, severity, and demographic distribution of mobility-related disabilities. Additionally, the large-scale, population-based nature of this national disability survey significantly enhanced the generalizability and relevance of the findings to the Saudi population, ensuring robust and contextually valid conclusions (Levin, 2019; Sedgwick, 2014).

  1. Given that chronic diseases had an important impact notably on women, what are the leading conditions impacting mobility and what strategies do you think could be proposed as to decrease the effects of these conditions?

-If available, it would be interesting to present (in a table) the most common chronic conditions that led to reduced mobility.

  1. You stated the inclusion criteria, but it is also important to state clearly the exclusion criteria for your sample.

Good luck!

Author Response

Reviewer 2

Dear authors,

The manuscript entitled “The Epidemiology of Mobility Difficulty in Saudi Arabia: National Estimates, Severity Levels, and Sociodemographic Differentials” is bringing into attention a major component of the dimension of disability.

While the topic is interesting and the quality of the manuscript is very good, there are some aspects that should be addressed:

Dear Colleague,

We appreciate your careful reading of our study and your thoughtful suggestions. All revisions appear in blue font, and each of your comments is answered point-by-point in the attached response document. Your feedback has materially improved the manuscript’s coherence and impact, and we thank you for your expertise.

Sincerely,

Authors

Please pay attention to typo, spaces, and overall aspect of the manuscript.

All checked and revised. Also please note that some formatting issues might be due to the journal template. 

Regarding the Methods section- Design, I recommend you add the following paragraph at the end, in the discussion section- Limitations: “However, as with all cross-sectional designs, limitations exist, such as the inability to establish causal relationships or temporality between risk factors and outcomes (Carlson & Morrison, 2023).

Thank you. This one and the previous one both moved to Limitations, following your suggestion and the other colleague as well.

Nevertheless, the descriptive value of this methodology outweighs these constraints, as the primary objective of the study was not to infer causation but rather to precisely describe the prevalence, severity, and demographic distribution of mobility-related disabilities. Additionally, the large-scale, population-based nature of this national disability survey significantly enhanced the generalizability and relevance of the findings to the Saudi population, ensuring robust and contextually valid conclusions (Levin, 2019; Sedgwick, 2014).”

 We also moved this to the Limitations to show strengths and limitation of our study together.  

Given that chronic diseases had an important impact notably on women, what are the leading conditions impacting mobility and what strategies do you think could be proposed as to decrease the effects of these conditions?

We added one paragraph in the discussion.

-If available, it would be interesting to present (in a table) the most common chronic conditions that led to reduced mobility.

Sorry, the data does not include specific types of diseases. We tried to get more details on this from the Ministry of Health, but we did not receive any response from them.

You stated the inclusion criteria, but it is also important to state clearly the exclusion criteria for your sample.

Thank you, added.

Good luck!

Round 2

Reviewer 1 Report

Comments and Suggestions for Authors

Congratulations.  I would like to leave two comments.

-My previous comment: “How many young adults, adults and elderly people were included in the study? Please include information about age. It is relevant to answer questions like “Considering Saudi context, was the prevalence of difficulty walking or climbing stairs more common only in older people? Please think about this."  The answer: "Thank you. Could you please refer to Table 4 Panel C, if you do not mind. The table shows the distribution of population with mobility difficulty by age and sex. Since it is an epidemiological study so we presented this as part of the results other than the method since the survey was inclusive of all ages.”

The table 4 Panel C does not show the distribution of population with mobility difficulty by age. The table shows the distribution of population with years of mobility difficulty. I ask the question to try to understand if disability is common or not among young people. If so, this would make the reality more serious. If the authors do not have this information, they can write a few words about it in the limitations section.

-Line 609: “5. Limitations”     or    “Strengths and Limitations”? 

 “Strengths and Limitations” could be an item of Discussion section such as 'Future Directions' and 'Implications'.  Please, think about this.

Author Response

Reviewer 1: Black text reviewer; different colours: author(s)

Comments and Suggestions for Authors

Congratulations.  I would like to leave two comments.

Dear Colleague,
Thank you so much for all your previous comments. We appreciate all you provided comments which helped in improving our manuscript. We completed also revising the manuscript based on the minor comments you provided this time.

Warm regards,

Authors

-My previous comment: “How many young adults, adults and elderly people were included in the study? Please include information about age. It is relevant to answer questions like “Considering Saudi context, was the prevalence of difficulty walking or climbing stairs more common only in older people? Please think about this."  The answer: "Thank you. Could you please refer to Table 4 Panel C, if you do not mind. The table shows the distribution of population with mobility difficulty by age and sex. Since it is an epidemiological study so we presented this as part of the results other than the method since the survey was inclusive of all ages.”

The table 4 Panel C does not show the distribution of population with mobility difficulty by age. The table shows the distribution of population with years of mobility difficulty. I ask the question to try to understand if disability is common or not among young people. If so, this would make the reality more serious. If the authors do not have this information, they can write a few words about it in the limitations section.

Please accept my deep apologies. You are totally right. I mixed between the duration and ages. We added these statements in the results and limitations:

Appendices 2 and 3 provide age-group distributions (counts and row-percentages) for Saudis reporting a single functional difficulty and for those reporting mobility difficulty plus at least one additional limitation, respectively, thereby enabling life-course comparisons of single- versus multiple-difficulty burdens.

More importantly, a key restriction of the public dataset is that age was tabulated only for the binary categories ‘with any difficulty’ versus ‘without difficulty’; because age was not cross-classified by specific difficulty type, we were unable to calculate age-specific prevalence of mobility difficulty per se or identify which age strata contribute most to the national burden—an analytic gap that future studies using individual-level micro-data will need to address.

-Line 609: “5. Limitations”     or    “Strengths and Limitations”?

Actually both, but following APA style, they just put this heading but include both. We added, strengths.

 “Strengths and Limitations” could be an item of Discussion section such as 'Future Directions' and 'Implications'.  Please, think about this.

You are right. We actually put all concluding parts as subsections of the discussion but apparently the production team wants them to be new sections. We corrected that now and we will leave it for them in case they want them to be separate sections.

Once again, thank you so much.

Warm regards,

Authors

Submission Date 29 June 2025

Date of this review 19 Jul 2025 18:47:09